# Laser-Based Mobile Visible Light Communication System

**DOI:** 10.3390/s24103086

**Published:** 2024-05-13

**Authors:** Yuqi Hou, Zhichong Wang, Zengxin Li, Junhui Hu, Chicheng Ma, Xiaoqian Wang, Liang Xia, Guangyi Liu, Jianyang Shi, Ziwei Li, Junwen Zhang, Nan Chi, Chao Shen

**Affiliations:** 1Key Laboratory for Information Science of Electromagnetic Waves (MoE), School of Information Science and Technology, Fudan University, Shanghai 200433, China; yqhou21@m.fudan.edu.cn (Y.H.); 23210720262@m.fudan.edu.cn (Z.W.); as508471157@163.com (Z.L.); jhhu21@m.fudan.edu.cn (J.H.); 20210720258@fudan.edu.cn (C.M.); jy_shi@fudan.edu.cn (J.S.); lizw@fudan.edu.cn (Z.L.); junwenzhang@fudan.edu.cn (J.Z.); nanchi@fudan.edu.cn (N.C.); 2ZGC Institute of Ubiquitous-X Innovation and Applications, Beijing 100876, China; wangxiaoqian@chinamobile.com (X.W.); xialiang@chinamobile.com (L.X.); 13701173393@139.com (G.L.); 3China Mobile Research Institute, Beijing 100053, China

**Keywords:** visible light communication, LiFi, laser lighting, semiconductor laser

## Abstract

Mobile visible light communication (VLC) is key for integrating lighting and communication applications in the 6G era, yet there exists a notable gap in experimental research on mobile VLC. In this study, we introduce a mobile VLC system and investigate the impact of mobility speed on communication performance. Leveraging a laser-based light transmitter with a wide coverage, we enable a light fidelity (LiFi) system with a mobile receiving end. The system is capable of supporting distances from 1 m to 4 m without a lens and could maintain a transmission rate of 500 Mbps. The transmission is stable at distances of 1 m and 2 m, but an increase in distance and speed introduces interference to the system, leading to a rise in the Bit Error Rate (BER). The mobile VLC experimental system provides a viable solution to the issue of mobile access in the integration of lighting and communication applications, establishing a solid practical foundation for future research.

## 1. Introduction

Traditional wireless communication technology has been widely used in people’s daily life, and the demand for wireless communication technology has grown exponentially, underscoring its importance in the modern world [1]. However, as demands for higher data rates, lower latency, increased capacity, and multiple connections grow, wireless communication technologies face significant bottlenecks and challenges. The emergence of visible light communication (VLC) promises a solution to these issues [2,3]. Unlike radio frequency communications, VLC offers abundant spectrum resources, superior interference resistance, and enhanced security. Importantly, VLC can utilize existing lighting infrastructure to provide dual functionality in illumination and communication, serving as an effective supplement to existing wireless networks, known as light fidelity (LiFi) [4,5]. Research on VLC over the past decades has made significant progress, achieving breakthroughs in communication rates through the optimization of transmitter and receiver devices, advancements in modulation and equalization algorithms, and the application of multiple-input multiple-output (MIMO) technology [6,7,8,9,10].

In laser-based VLC systems, laser diodes (LDs) present a promising avenue due to their high modulation bandwidth and output power, indicating potential for high-speed and long-distance communications [11,12]. However, the narrow beam and concentrated spot of LDs necessitate precise alignment for high-capacity transmission, which severely limits the mobility of communication systems. Mobility, a fundamental advantage of wireless services, provides users with flexibility and convenience, enabling connections while on the move. In the age of mobile internet, there is a universal expectation for ubiquitous communication capabilities, requiring communication systems that accommodate mobility. Actually, the vast majority of envisioned applications for VLC require implementation in mobile scenarios. For instance, smart homes, smart cities, and factory floor communication, among others. In short-range scenarios, the necessity of mobile VLC is evident in electromagnetic-sensitive environments such as hospitals or where current mobile technologies struggle to provide coverage. For instance, on trains or airplanes, VLC technology enables passengers to quickly establish ad hoc networks for information exchange and local social networking applications. In emergency situations, emergency lights or exit signs can communicate with victims’ portable devices, providing the shortest paths to safety exits. In addition, the study of mobile VLC is also important for underwater communications, especially in the presence of turbulence [13]. Given that most envisioned applications of VLC operate in mobile scenarios, there is an urgent need to explore mobile VLC systems. We envisioned an indoor LiFi application scenario for mobile VLC, as depicted in Figure 1. People’s mobile phones, computers, and other mobile devices can wirelessly connect through indoor lighting and connect to control electronic devices such as printers and projectors.

To ensure the effective maintenance of a VLC link during mobility, it is first imperative to guarantee that the light source not only illuminates within a certain range, but can also support communication. Thus, the study of wide field-of-view (FOV) VLC holds significant importance. Lasers, characterized by their narrow divergence angles, have traditionally been utilized in point-to-point communications. Consequently, there is a need to expand the illumination range of laser beams and explore large-angle receiving solutions. Some researchers have delved into this area in recent years. Chun et al. innovatively extended the coverage range of a light source to a full angle of 76° through the integration of Micro-Electro-Mechanical Systems’ steering mirrors and angle amplification modules. Laser-based white light consists of red, green, blue, and violet LDs four color multiplexing. The system operates at ultrahigh data rates of more than 35 Gb/s with a coverage area of 39 m^2^ at a link distance of 4 m [14]. The fused fiber-optic tapers designed by Alkhazragi et al. offer a wide-FOV laser reception solution. The optical detector offers a high efficiency, broad modulation bandwidth, indefinite stability, and operates over a wide range of wavelengths. Between −32.5° and 32.5°, the system can support 1 Gbit/s using on–off keying (OOK) based on a 642-nm laser [15]. Yu et al. experimentally compared the performances of an engineered diffuser (ED) and ground glass diffuser (GGD), and verified the diffusion effect of a green LD through a diffuser, enabling transmission at a rate of 1 Gbps within a half-angle range of 12° in a 2 m air channel. The light source can also support communication in a 2 m water channel [16]. In our previous work, by exciting fluorescent ceramics with an inclined blue laser, we achieved an FOV of 34°. At a 1 m distance, the highest data rate can exceed 3.7 Gbps at the center, and the overall transmitting rate is beyond 3 Gbps within a 34° range [17]. By placing an engineered diffuser at the transmitter and using a Silicon Photomultiplier (SiPM) for reception, this angle can be further expanded to 180° [18]. These wide-FOV laser-based VLC solutions provided numerous references for this work.

The realization of wide-FOV communication provides solutions for the transmitting and receiving devices in mobile VLC systems, and research has also expanded into mobile VLC channel and system models. Kinoshita et al. studied the variations in mobile channels between cars through image optical flow, which is the coordinate changes of the VLC transmitter in images during movement, and established a motion model for theoretical research. Simulations based on proposed motion models indicated motion characteristics for infrastructure-to-vehicle VLC, vehicle-to-infrastructure VLC, and vehicle-to-vehicle VLC [19]. You et al. conducted vehicle-to-infrastructure channel estimation based on decision feedback. Through simulation under this method, an effective visible light communication link can be established when the vehicle’s moving speed is 15 m/s [20]. Liu et al. developed a model for high-speed train communication, exploring the impact of the Doppler effect on this type of high-speed mobile VLC model. Adaptive modulation schemes were investigated for both the single-input single-output (SISO) system and the MIMO system under the high-speed mobile VLC model [21]. Qin et al. established a noise model for vehicular VLC to obtain the optimal error performance [22]. Zhu et al. established a simulation model for mobile vehicles and multiple streetlights to investigate the switching strategy between adjacent streetlights, aiming to prevent signal interruption caused by light source switching. Their receiving signal power (RSP) handover scheme can ensure normal communication between vehicles and lights with a low cost, increased signal-to-noise ratio (SNR), and reduced bit error rate (BER) [23]. Hussein et al. improved the system degrees of freedom by addressing channel crosstalk with delay-adaptive techniques through simulation studies [24]. These works provide a solid theoretical foundation for research on mobile VLC, thoroughly demonstrating the potential of VLC applications in mobile environments, while there is currently a significant lack of research on experimental systems for mobile VLC. The first demonstration was proposed by Hong et al. in 2016, and they suggested orthogonal circulant matrix transform (OCT) precoding to be a promising way to deal with the high package loss rate [25]. In 2018, Shi et al. put forward another experiment, where the transmitting rate was only 4 kbps [26]. For clarity, we summarize previous research on mobile VLC in Table 1.

In this study, we designed an experimental platform for mobile VLC and conducted a series of experiments to assess its performance. Throughout these experiments, we transmitted OOK signals at a rate of 500 Mbps. The lens-free system demonstrated a stable link performance at distances of one meter and two meters, showing adaptability to mobility-induced perturbations. However, beginning at a distance of three meters, increases in both distance and speed contributed to a larger fluctuation range in the BER. Despite these challenges, at a distance of four meters, the system was still capable of supporting a transmission rate of 500 Mbps at a speed of 0.48 m/s. This choice of speed takes into account the mobile communication needs of people indoors such as offices and classrooms. Though distance and mobility impact the system’s performance, it maintained a considerable degree of reliability and efficiency.

## 2. Experiment Setup

To achieve a wide FOV for a white-light laser source, we employed a diffuser plate to scatter a commercial high-power laser array, transforming it into a non-directional point light source. Additionally, the use of a plano-convex lens further enhanced the uniformity and stability of the light beam. The light source generated white light through the fluorescence conversion of 452 nm blue light. The fluorescent material absorbed the blue light and converted it into a broad wavelength light, the spectrum of which is shown in Figure 2a, while the chromaticity coordinates are presented in Figure 2b. The light source exhibited a color temperature of 8318 K and a color rendering index (CRI) of 67.6, both of which demonstrate a commendable current stability. More details about this light source and its illumination and communication performance can be found in reference [27].

The optical power distribution of the light source is shown in Figure 3. To obtain accurate optical power measurements, we used a bandpass filter with a wavelength of 450 nm before measuring its optical power. The power meter we used (LBTEK PPM200S-20-USB, Shenzhen, China) had a maximum incident beam diameter of 10 mm. Clearly, as the distance increased, the optical power gradually decreased. At the same distance, the power also attenuated with an increasing angle, with the maximum power at the central angle. When the distance was one meter, the optical power at the central angle was 368 μW, maintaining at a relatively high power level within a certain angle range, then dropping rapidly. Within a range of ±10 degrees, the optical power remained above 300 μW. By 20 degrees, the power had already decreased below 100 μW. The overall trend at two meters was similar to that at one meter, with the maximum optical power at the center being 101.1 μW. Within a range of ±10 degrees horizontally, the power remained above 80 μW, and beyond 10 degrees, the power attenuated rapidly. Centered on the orientation of the light source, the emission range of the light extended approximately 22 degrees to both the left and right. This characteristic offers a promising way for developing high-quality, wide-FOV laser illumination sources. Such a directional spread ensures that the illumination can cover a wide area while maintaining a focused intensity to support high-speed communication.

To facilitate communication measurements during mobility, we designed and customized a set of electric rails and a control system specifically for mobile VLC, as depicted in Figure 4.

The track was 1.2 m in length, with a safety margin of 0.2 m at each end, resulting in a general movement range of 0.8 m along the center of the track, corresponding to coordinates of 0–0.8 m. The rotation of the motor drove the moving platform to slide on the track via a synchronous belt. The pitch of the synchronous belt was 162 mm per revolution, meaning that when the speed was set to 1 S/r, the mobile platform moved forward 162 mm in one second. A control box was positioned on the left side of the track, enabling the movement of the platform at different predetermined speeds.

The VLC experimental system, including the communication equipment, is depicted in Figure 5. In Figure 5a, the perspective is from the transmitting end at one meter. As our light source provided broad coverage, it can be observed from the figure that the entire track was within the coverage area of the light spot. Therefore, theoretically, the receiving end could transmit via the visible light link when moving along the track. The receiver, an SiPM, was placed on the moving platform to achieve signal transmission during motion, as illustrated in Figure 5b. The setup of the mobile VLC system is shown in Figure 5c. The system’s transmitter was located on the left, mounted on a movable cart to change the distance between the transmitter and receiver, thereby allowing the study of mobile communication performance over varying distances. On the right side was the system’s receiver, with the mobile communication track placed on a desktop with the receiving detector on it.

The OOK signal was generated by an arbitrary waveform generator (Keysight M8190A, Santa Rosa, CA, USA), and its amplitude was adjusted through an amplifier (Mini-Circuits ZHL-5W-1+, New York, NY, USA) and an attenuator (Alf Communications Key-Press Attenuator kT2.5-30/1S-2S, Shenzhen, China). Subsequently, the signal was coupled with a direct current to drive the light source to emit light, using a bias tee (Mini-Circuits ZFBT-282-1.5A+, New York, NY, USA). After traversing the mobile-free space channel, the SiPM collected the optical signal and converted it into an electrical signal, which was then forwarded to an oscilloscope for sampling, demodulation, and BER calculation.

It is noteworthy that we utilized a lens-free receiver, which possesses considerable advantages in practical applications. The absence of lenses in the design simplifies the optical architecture, reducing the complexity and potentially lowering the cost of the system. Furthermore, this configuration allows for a more compact and lightweight design, making it more suitable for integration into mobile devices and applications where space and weight are critical constraints.

## 3. Results and Discussion

Considering the bandwidth limitations of the components within the system, we transmitted an OOK signal at a rate of 500 Mbps. Initially, we examined the variations in the BER when the SiPM remained stationary at different positions along the track, as illustrated in Figure 6. The BER values represented in the graph are the averages of at least twenty measurements taken at specific distances and positions, with different color data points indicating the communication performances at varying distances between the transmitter and receiver. The left, center, and right positions correspond to track coordinates of 0 m, 0.4 m, and 0.8 m, respectively.

Firstly, when observing the variation in the BER with distance, it generally followed the pattern that the BER increased as the optical transmission distance lengthened. Taking the example of the BER with the receiver positioned in the middle, the BER was 1.4 × 10^−4^ at one meter, 1.7 × 10^−4^ at two meters, 2.8 × 10^−4^ at three meters, and 6.9 × 10^−4^ at four meters. This was owing to the reduced received optical power with a longer transmission distance. An exception occurred at distances of 1 m at both ends, where the BER at 1 m was higher than that at 2 m. At two meters, the BER at both ends was only 1.7 × 10^−4^, whereas at one meter, it was 2.4 × 10^−4^. The analysis suggests that this abnormal phenomenon can be attributed to the 22-degree FOV of the light source. Consequently, at a distance of 1 m, the diameter of the light spot was around 0.8 m, meaning that the measurement points on either side of this distance were already at the edges of the light emitted by the source. The quality of the light at the very edges deteriorated, and with rapid power attenuation, this led to an increase in the BER.

Upon observing the variation in the BER at the same distance but different positions, it was evident that the BER at the central position was lower than that at the lateral positions. The difference became nearly negligible at 2 m, indicating that, at a distance of 2 m, the edge light intensity was already sufficient to support data transmission. Being directly in front of the light source may potentially have saturated the SiPM, hence, the BER at the central position did not significantly decrease further. At 1 m, as mentioned previously, due to being at the edges of the FOV of the light source, the BER on the sides noticeably increased again.

Following the analysis of stationary conditions, we proceeded to compare the performance of the receiver when in motion to when it was stationary. The stationary data include measurements taken at fixed positions at track coordinates of 0 m, 0.4 m, and 0.8 m. The dynamic data were sampled at random positions during motion, including various motion speeds. Taking into account the influence of a one-meter coverage, the experiment contrasts the BER distribution at stationary and mobile states for distances of two meters, three meters, and four meters, as depicted in Figure 7.

At a distance of 2 m, Figure 7a reveals that the distribution of the BER in both stationary and mobile VLC systems was essentially consistent, varying from 1.0 × 10^−4^ to 3.5 × 10^−4^, with the BER range in mobile VLC slightly exceeding that of stationary VLC. This indicates that, at this juncture, the movement of the receiver did not significantly impact the system, allowing for stable transmission under any condition. However, as the distance extended to 3 m and 4 m, the BER was significantly higher, and there were a small number of data that were far off, as illustrated in Figure 7b,c. Interestingly, data deviation occurred in both stationary and mobile contexts, with comparable frequencies. As shown in Figure 7b, whether in stationary and mobile conditions, the majority of the BER data points clustered around 5.0 × 10^−4^. However, there were a few points that appeared at values exceeding 1.0 × 10^−3^. We tend to believe that this was brought about by having environmental perturbations and judgment error rather than the movement of the receiving end. Consequently, we excluded these environmental factors and focused on the boxed portion of data for further observation.

As illustrated in Figure 7d,e, an increase in distance introduced instability to the system, gradually widening the BER distribution range. Moreover, the impact of the mobile receiver on the system performance became more pronounced, with the BER distribution under mobile VLC conditions being noticeably more dispersed. At three meters in stationary conditions, the BER fluctuated within a range from approximately 2.0 × 10^−4^ to 6.0 × 10^−4^, while in mobile conditions, this extended to a range from 1.5 × 10^−4^ to 7.0 × 10^−4^. At this point, the moving receiver had already exerted a certain influence on the system’s performance, resulting in a less stable distribution of the BER. At four meters, the distinction between stationary and mobile systems became more pronounced. In stationary conditions, the distribution of the BER was relatively concentrated, ranging from around 5.0 × 10^−4^ to 1.5 × 10^−3^. However, in mobile conditions, the BER was more dispersed, with distributions ranging from 5.0 × 10^−4^ to 3.5 × 10^−3^. To observe this more clearly, the experiment calculated the standard deviation of the BER from 1 m to 4 m for both stationary and mobile VLC, as shown in Figure 7f. At a distance of 1 m, the significant variation in the BER across different stationary positions resulted in a larger standard deviation for the stationary BER at this distance, exceeding that of the mobile condition. At distances of 2 m, 3 m, and 4 m, the standard deviation for mobile conditions consistently surpassed the level in stationary conditions. Furthermore, as the distance increased, both the stationary and dynamic standard deviations gradually rose, with the disparity between them continuously expanding.

Lastly, we further assessed the impact of the receiver’s moving speed on the system performance. Under varying distances from 1 m to 4 m, the system’s BER at different speeds (excluding environmental factors) is depicted in Figure 8. 1 S/r represents that the rotor completes one round per second, resulting in the mobile platform moving 162 mm. Our selected range for rotation speed ranged from 0.1 S/r to 3.0 S/r, with a corresponding speed of 0.48 m/s at 3.0 S/r. We opted to test at a speed of 0.48 m/s, taking into account the typical pace of movement for individuals within indoor settings. By simulating this common speed, we aimed to evaluate the performance and feasibility of our system under conditions reflective of everyday movement within indoor spaces. We tested about 40 sets of data at 1 m, 30 sets at 2 m, and 20 sets at 3 m and 4 m at each speed. The number of tests remained consistent across different velocities at the same distance. Then, we sorted the data obtained from the experiment from small to large to facilitate our subsequent drawing. After sorting, we used different colors to represent the relative size of the data. The blue part represents the smaller part of the BER, and the red part represents the larger part of the BER. Each small color block in the picture are the data obtained from one test.

According to Figure 8, when transmitting data at 500 Mbps over distances of 1 m, the distribution of bit error rates for speeds ranging from 0.1 S/r to 3 S/r was concentrated between 5.7 × 10^−5^ and 2.88 × 10^−4^. At a 2 m distance, the distribution was concentrated between 1.0 × 10^−4^ and 3.46 × 10^−4^, while at a 3 m distance, it ranged from 1.58 × 10^−4^ to 5.76 × 10^−4^. For the 4 m distance, the distribution was concentrated between 3.8 × 10^−4^ and 3.54 × 10^−3^. While there was some overlap between these data points, their lower and upper bounds consistently exhibited a well-defined monotonic trend with distance. Specifically, as the distance increased, the error rate also tended to increase. The overlap might have been caused by the SiPM utilized as a detector in the experiment, which was highly sensitive. Even minor disturbances in the channel can lead to variations in the detection results. As evident from the static and dynamic BER distributions in Figure 7, the BER fluctuated within a certain range, even during stationary conditions. Therefore, it was inevitable that there would be fluctuations in the BER during the movement of the receiver on the track. Rather than comparing one single BER, we focused on the overall trend in the distribution range.

For the 1 m and 2 m cases, as shown in Figure 8a,b, the BER across different speeds did not display a clear change, suggesting that the system’s performance remained highly stable, even in the face of rapid movement. However, upon extending the transmission distance to 3 m in Figure 8c, it became apparent that, at lower speeds, the blue segment was longer and the red segment was shorter. At higher speeds, the length of the blue segment decreased while the red segment lengthened, indicating that the instability introduced by rapid movement resulted in an increase in the BER. This pattern in Figure 8d became even more evident at a transmission distance of 4 m. At speeds below 1.0 S/r, almost all bit error rates fell within the blue range. As the speed increased, the upper limit of the BER gradually rose, and by 3.0 S/r, nearly half of the BERs were in the red range, demonstrating significant variability in the system performance. However, it is noteworthy that the BER remained below 3.8 × 10^−3^ throughout, indicating an effective transmitting system.

We averaged the BERs measured at a certain distance for various speeds and plotted them in Figure 9. It is evident that, at a fixed distance, the BER increased gradually with the increase in speed, particularly noticeable at distances of 3 m and 4 m. On the other hand, when maintaining consistent speed parameters, as the distance between the transmitter and the receiver became larger, the BER also showed a trend of becoming larger. Overall, as the distance and speed increased gradually, the experimentally measured BER also increased accordingly. Conversely, when the distance was short and the speed was low, the BER tended to be at its lowest. This observed trend aligns well with common knowledge.

## 4. Conclusions and Discussion

In this work, we present the design of a mobile VLC system architecture and conduct a comprehensive experimental analysis of its data transmission performance. Through experimental design and detailed performance evaluation, the results demonstrate robustness at transmission distances of one meter and two meters, showing that the mobility of the receiver has a minimal impact on the system performance. However, as the transmission distance increases, the receiver mobility will impact the system performance and intensify with an increased mobility speed. At a transmission distance of four meters, the system is capable of supporting a transmission of 500 Mbps at a speed of 0.48 m/s.

The experiments validate the effectiveness and reliability of the mobile laser-based VLC system under various operational conditions and provide valuable performance references for the application of mobile VLC technology in real-world environments. With ongoing development and optimization, the potential applications of mobile VLC technology in scenarios such as vehicular communication, connectivity among mobile devices, and indoor positioning are expected to expand significantly. Mobile VLC technology is poised to play an increasingly important role in the construction of intelligent communication networks. 

In further research on VLC in mobile scenarios, several promising directions can be pursued. Firstly, integrating advanced equalization and modulation techniques tailored specifically for mobile environments can significantly enhance the robustness and reliability of VLC systems [28,29]. This could involve the development of adaptive algorithms capable of dynamically adjusting parameters in response to changes in ambient lighting conditions and mobility patterns. Secondly, exploring the potential of MIMO technology in VLC systems for mobile applications holds considerable promise. By leveraging multiple transmit and receive antennas, MIMO can improve data rates, extend coverage, and enhance the overall performance of VLC systems in dynamic mobile environments [8,30]. Furthermore, investigating novel hybrid VLC–radio frequency (RF) communication schemes could offer synergistic benefits, combining the strengths of both technologies to achieve seamless connectivity and an improved quality of service for mobile users [31,32]. This hybrid approach could involve intelligent handover mechanisms between VLC and RF networks, ensuring uninterrupted communication and optimal resource utilization. Moreover, exploring innovative VLC-enabled applications beyond traditional communication, such as indoor positioning, augmented reality, and vehicular communication, could open up new avenues for research and practical implementation in diverse mobile scenarios [33,34,35]. Overall, by addressing these research directions, we can unlock the full potential of VLC in mobile environments, paving the way for the development of high-performance, ubiquitous communication systems that meet the evolving needs of modern mobile users.

## Figures and Tables

**Figure 1 sensors-24-03086-f001:**
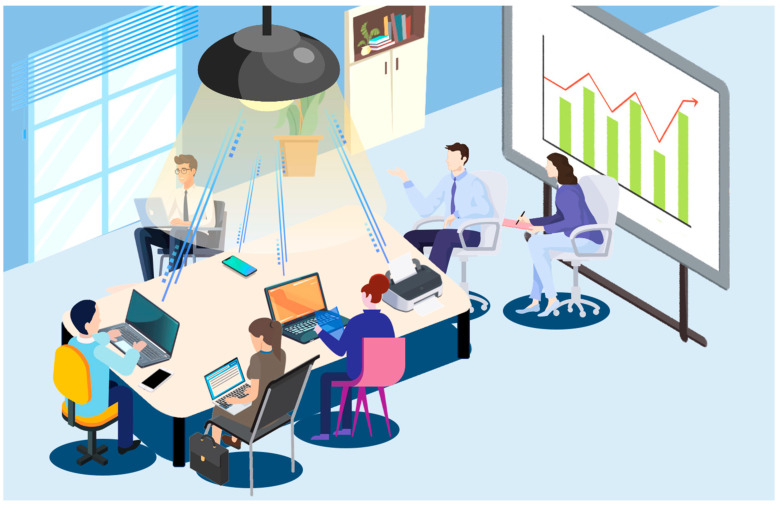
Indoor mobile VLC application scenario.

**Figure 2 sensors-24-03086-f002:**
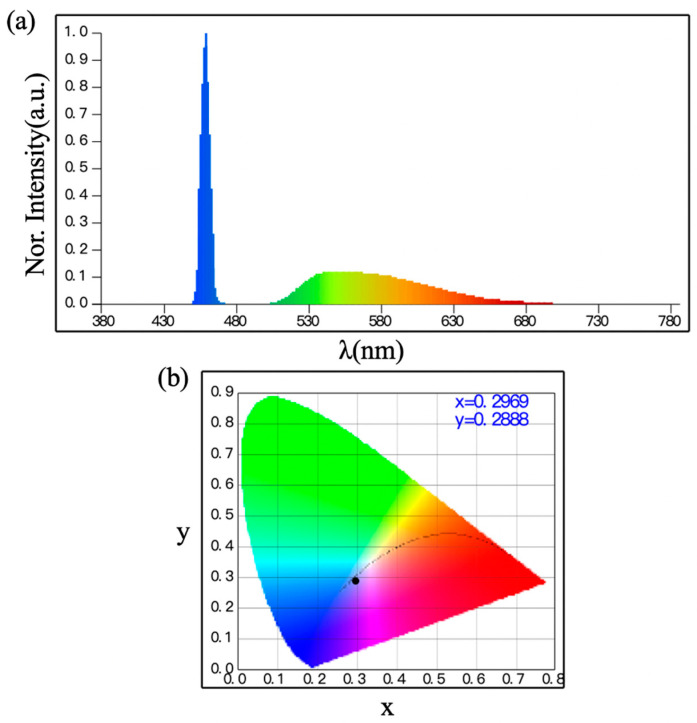
(**a**) Spectrum of the laser-based white light we employ in this work. (**b**) Chromaticity coordinates of the laser-based white light. The emission peak is located at 452 nm, the coordinates are x = 0.2969, y = 0.2888.

**Figure 3 sensors-24-03086-f003:**
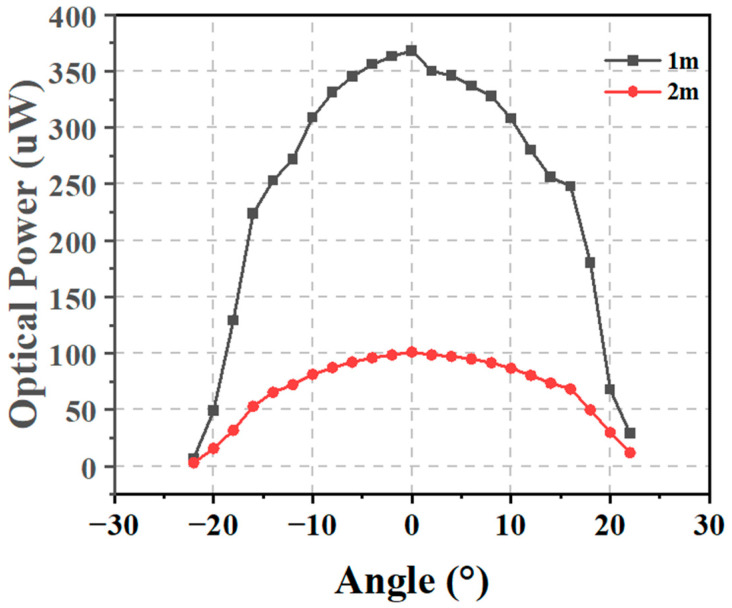
Optical power distribution of the light source after passing through a blue light filter at 1 m and 2 m.

**Figure 4 sensors-24-03086-f004:**
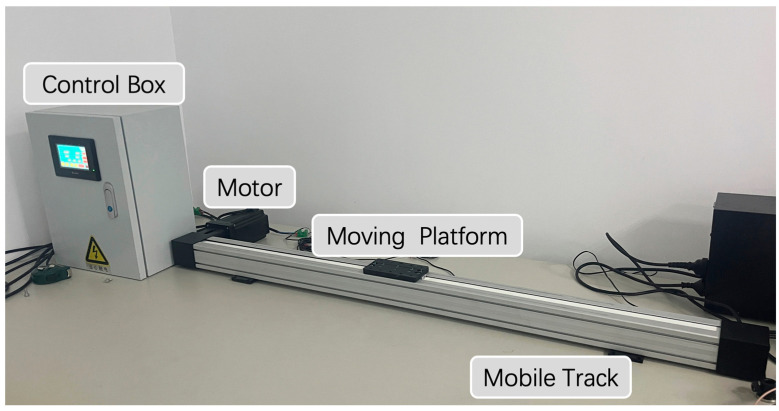
Mobile track and controlling equipment.

**Figure 5 sensors-24-03086-f005:**
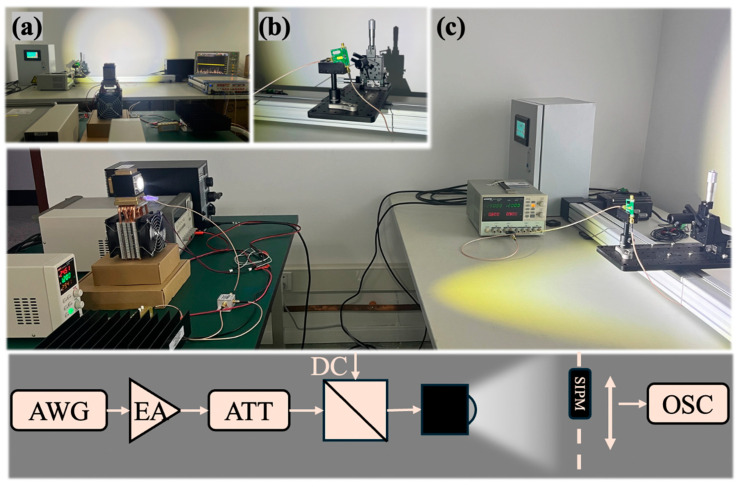
Experimental setup and flowchart of the system, including: (**a**) Light source shining on mobile VLC track. (**b**) Receiving end SiPM on mobile platforms. (**c**) Overall VLC system.

**Figure 6 sensors-24-03086-f006:**
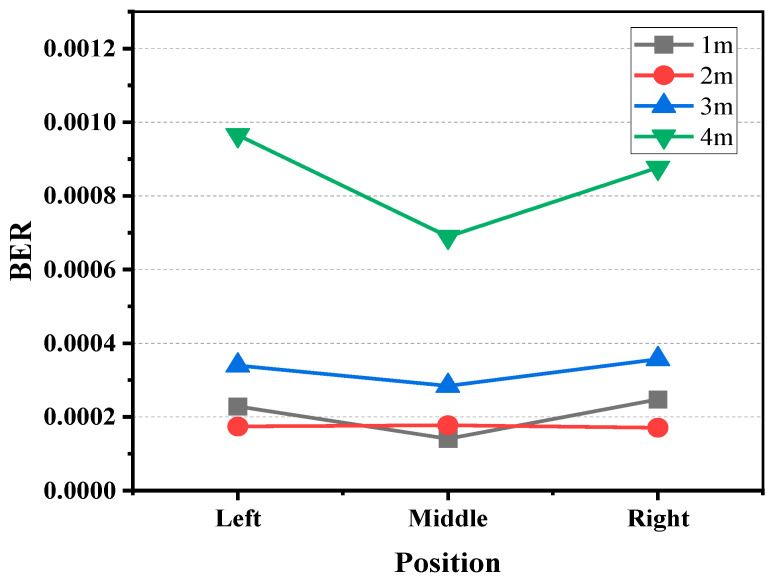
Comparison of BER at different reception locations for 500 Mbps data sent at rest at 1 m, 2 m, 3 m, and 4 m.

**Figure 7 sensors-24-03086-f007:**
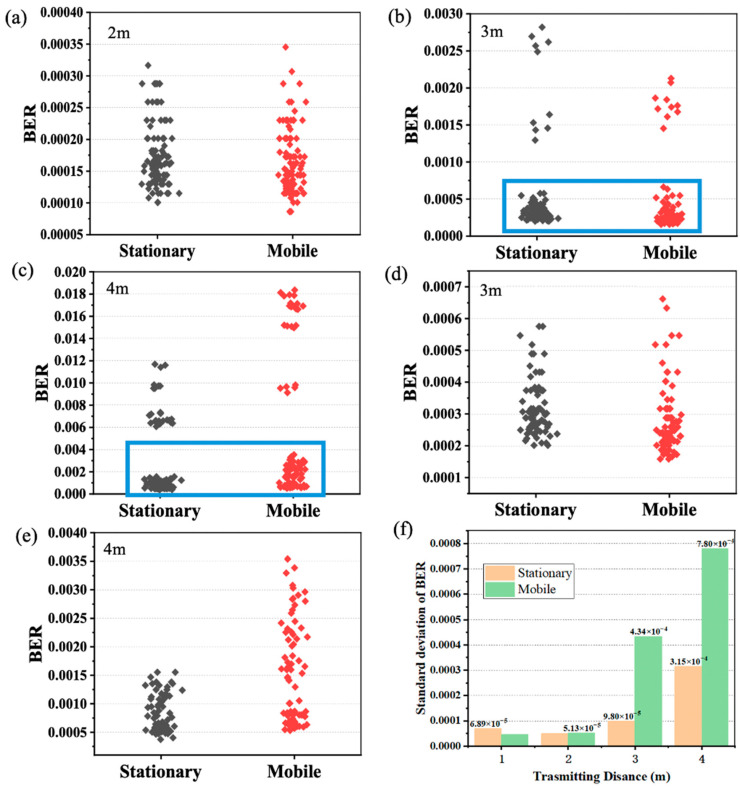
(**a**–**c**) distribution of stationary and mobile VLC BER at 2 m, 3 m, and 4 m. (**d**,**e**) partial enlargement of the boxes in (**b**,**c**) at 3 m and 4 m. (**f**) Comparison of standard deviation of BER between stationary and mobile VLC.

**Figure 8 sensors-24-03086-f008:**
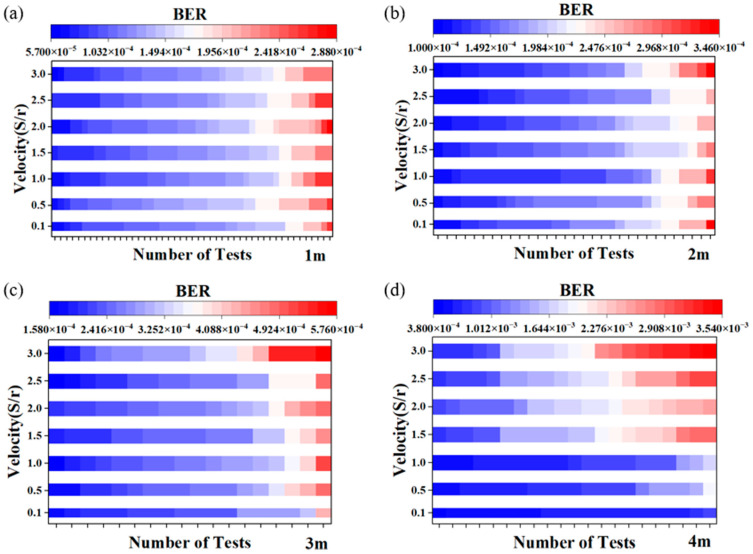
(**a**–**d**) BER of the mobile VLC system for different receiver moving speeds at 1 m, 2 m, 3 m, and 4 m.

**Figure 9 sensors-24-03086-f009:**
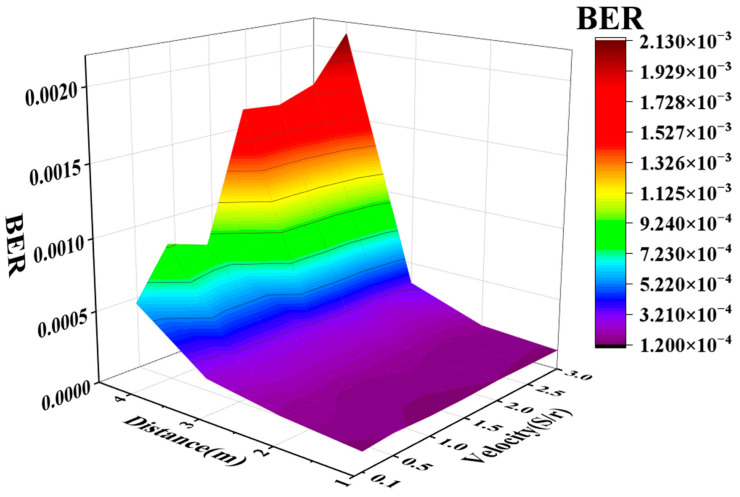
The variation in mean BER with position and velocity.

**Table 1 sensors-24-03086-t001:** Previous works on mobile VLC.

Light Source	Type	Technology	Moving Velocity	Transmitting Rate	Distance	Organization	Year	Citation
LED car light	Simulation	Pinhole camera model in image sensor-based VLC	30 km/h	N/A	30 m	Nagoya University	2014	[19]
Multiple LED streetlamps	Simulation	Decision feedback channel estimation and MIMO	15 m/s	199 Mbps	1.5 m	Nanjing University of Posts and Telecommunications	2020	[20]
Multiple LEDs	Simulation	Adaptive modulation	300 km/h	N/A	1 m	Southeast University	2021	[21]
LED streetlight	Simulation	Allan variance and adaptive Gaussian mixture-model-based noise modeling	50–70 km/h	N/A	3–13 m	Xi’an University of Technology	2023	[22]
Multiple LED streetlights	Simulation	Multiple transmitter RSP handover	10–25 m/s	N/A	3 m	Jiangsu University	2014	[23]
RGB-LDs	Simulation	Delay adaptation technique	1 m/s	N/A	3 m	University of Leeds	2015	[24]
Blue LD	Experiment	OCT precoding	20 cm/s	300 Mbps	1 m	The Chinese University of Hong Kong	2016	[25]
White-light LED	Experiment	Dynamic column matrix selection algorithm	20–100 cm/s	4.08 kbps	60 cm	Hunan University	2018	[26]

## Data Availability

The raw data supporting the conclusions of this article will be made available by the authors on request.

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
