# Peer review of "Laser-Based Mobile Visible Light Communication System"

_sensors, 2024, doi:10.3390/s24103086_

Round 1

Reviewer 1 Report

Comments and Suggestions for Authors

The authors proposed a design for a mobile visible light communication (VLC) system architecture, and experimentally verified the effectiveness and reliability of the mobile laser VLC system under various operating conditions, providing valuable performance references for the practical application of mobile VLC technology. The reviewers identified several issues that need to be addressed in the paper:

1. Whether the light source used in the experiment is a commercial laser or a homemade laser? It is recommended to clarify in the text.

2. In line 115, the paragraph lacks a period at the end. Please review the entire content for editing.

3. In Figure 6, please explain why the error rate distribution for transmission distances of 3m and 4m is better than that for transmission distances of 1m and 2m when the rotational speed is in the range of 0.1-1.0?

4. It is suggested to unify the format of the reference list and add several cited papers.

Comments on the Quality of English Language

Minor editing of English language required.

Reviewer 2 Report

Comments and Suggestions for Authors

Relevance: This work is quite relevant towards advancing visible light based wireless communication systems using Lasers instead of conventional LEDs.  

Novelty: The work does uses techniques to improve the Field of View which is a limitation for alignment using Laser based wireless communication systems. The technique includes using a diffuser plate along with generating white light with fluorescence conversion. The experimental setups and results presented follow from this idea. 

Literature Review: The literature review is presented well. 

Analysis: Experiments and analysis to measure BER under different stationery and mobiles conditions are presented. Details of the experimental setup, measurement setup and BERT results are presented very well. Noteworthy are result showing the receiver speed, which are very useful. In cases where results are counterintuitive, such as Fig. 5, additional analysis was performed to clarify the inconsistencies.   

Quality of Plots: The quality of some plots needs to be improved. For example: The x-axis on Fig. 6 for all sub-plots are not clear.  

Comments on the Quality of English Language

The quality of English is satisfactory. There are few corrections that I want to suggest. 

1) Line 111: Should be centre

2) Line 201: Should be Fig. 6

3) Line 222: Should be conducts

Reviewer 3 Report

Comments and Suggestions for Authors

This manuscript reported an experimental platform for mobile VLC and conducted a series of experiments to assess its performance. Throughout these experiments, we transmitted on-off keying (OOK) signals at a rate of 500 Mbps. The lens free system demon-81 strated stable link performance at distances of 1 meter and 2 meters, showing adaptability to mobility-induced perturbations. It is very interesting. What I am concerned about is its application scenario? Is it necessary to communicate in this way for 1 meter and 2 meters?

Comments on the Quality of English Language

Minor editing of English language required.
